# V-Shaped Toothed Roller Cotton Stalk Puller: Numerical Modeling and Field-Test Validation

Zhenwei Wang [1], Weisong Zhao [1], Jingjing Fu [1], Hu Xie [1], Yinping Zhang [2,*] and Mingjiang Chen [1,*]

[1] Nanjing Institute of Agricultural Mechanization, Ministry of Agriculture and Rural Affairs, Nanjing 210014, China; wangzhenwei@caas.cn (Z.W.); wszhao77@163.com (W.Z.); tutujing12@163.com (J.F.); tmjamexh@163.com (H.X.)

[2] School of Agricultural and Food Science, Shandong University of Technology, Zibo 255000, China

[*] Correspondence: zhangyinping@sdut.edu.cn (Y.Z.); chenmingjiang@caas.cn (M.C.)

**Abstract:** The V-shaped toothed roller cotton stalk puller has a low removal ratio and weak pulling effect. Hence, we constructed a simplified mathematical model of the V-shaped tooth roller stalk puller based on elastic collision theory and simple beam theory and conducted a mechanical analysis based on this model to explore the causes of pulling errors and fractures. Specifically, the V-shaped tooth plates of the machine were optimized in an orthogonal experiment with the rotational speed, cogging angle, and ground clearance as the influencing factors, and the removal ratio as the evaluation index. This experiment was designed to enable analysis of the physical characteristics of cotton stalks, and the forces applied during the pulling process. Additionally, a V-shaped toothed roller-type stalk-pulling test bench was constructed. The results revealed that, unlike the cogging angle, the ground clearance significantly affected the removal ratio. Furthermore, the highest removal ratio (i.e., 97%) was achieved when the ground clearance was −20 mm, the rotational speed was 300 rpm, and the cogging angle was 32.5°. An $L_9$ ($3^4$) orthogonal field experiment was also conducted with the rotational speed, cogging angle, and ground clearance as the influencing factors to investigate their respective influences on the stalk removal ratio. The results revealed that the ground clearance most significantly influenced the ratio, followed by the rotational speed, and cogging angle. The ground clearance and rotational speed of the V-shaped toothed roller were each found to significantly influence the ratio. Furthermore, a ground clearance of −20 mm, rotational speed of 300 r/min, and cogging angle of 25° yielded an average removal ratio of 98.27%. Through this research, the mechanism of toothed roller stalk pulling is further deepened and the toothed series stalk pulling technology provides theoretical support.

**Keywords:** V-shaped tooth roller; removal ratio; orthogonal experiment; structure design





## 1. Introduction

Cotton stalk is a byproduct of cotton that can be utilized as a high-quality renewable resource. China is a large cotton-producing country. In 2020, the national cotton planting area was 3.1699 million square hectares [1,2]. More than 40% of the world's cotton is produced in China, and this area produced a total output of $3.05 \times 10^7$ t of cotton stalks [3,4]. However, only approximately one-tenth of this total output was subsequently utilized. The lack of effective cotton stalk harvest machinery is one of the main reasons for this low utilization rate [5,6].

The core component of cotton stalk harvester machinery is the stalk-pulling structure, the performance of which directly influences stalk-pulling efficiency. Thus, to increase the cotton stalk removal ratio, many researchers have investigated the characteristics of cotton stalks, the mechanism of the stalk-pulling structures, and the dynamic characteristics of stalk pulling. Regarding the mechanical properties and pull-out characteristics of cotton stalks, Chen et al. [7,8] examined how the cotton stalk pulling force changed in the field

between autumn and the following spring. They obtained the following results: (1) the tensile failure load was 10.9 to 29.8 times larger than the bending failure load and (2) the pulling resistance of the cotton stalk is significantly influenced by the diameter of the cotton stalk root, the soil hardness, and the harvest time. Using a pull-out force measuring device, Li et al. found that the cotton stalk pulling angle significantly affects the pulling force [9,10]. Additionally, the findings reported by Demian T. F. et al. indicate that, within a certain cotton stalk height range, a higher pulling height mandates a larger pulling force [11]. Currently, there are two general types of straw-pulling structures: the row-controlled pulling structure, which includes the tooth-disc type, double-roller type, and chain-clamp type; and the non-row-controlled structure, which includes the knife-roller type and V-shaped toothed roller type. To date, non-row-controlled structures have been preferred over row-controlled structures because they have advantages such as lower operational requirements for the driver and higher working efficiency [12–17].

Among the non-row-controlled stalk-pulling structures, the knife-roller type has a high removal ratio; however, because it penetrates the soil during operation, it is associated with high energy consumption and problems related to residual film and mud, which hinders the follow-up work. The results of theoretical research have shown that, although it is still in an exploratory phase, the V-shaped toothed roller type not only has the advantage of a high removal ratio commonly associated with the knife-roller type but also the advantage of low energy consumption [17]. Tang et al. designed an all-in-one V-shaped toothed roller machine by integrating the functions of cotton stalk pull out, soil clearance, transportation, chopping, and collection; this design yielded a high extraction rate [12]. However, an in-depth theoretical analysis of the V-shaped toothed roller was not performed. The Nanjing Institute of Agricultural Mechanization, Ministry of Agriculture and Rural Affairs, and Binzhou Agricultural Mechanization Research Institute jointly developed a V-shaped toothed roller-type cotton stalk harvester that could perform the functions of stalk pull-out, soil clearance, chopping, and bundling [18–20]. This collaboration resulted in the first application of an integrated elastic collision and simple beam theory to stalk pulling [21]. Thereafter, He et al. and Dai et al. used this theory to study and test other types of stalk-pulling machines [22,23]. However, to date, how the forces applied to V-shaped toothed rollers vary as the stalks are extracted remains to be unknown, and the reasons for cotton stalk fracture and non-pulled stalks lack clear theoretical explanations.

A small-scale test bench is designed in this work, which is driven by hydraulic pressure to pull the stalk roller, and the motor drives the reel and is equipped with torque, speed sensors, and other components that can control the motion parameters very accurately so as to facilitate the research of the experiment. In order to elucidate the operational stress variation profile for V-shaped toothed roller stalk-pulling machines and explore the mechanisms of cotton stalk pull-out, fracture, and missed extraction, elastic collision and simple beam theories were applied to develop a comprehensive stalk-pulling theory. The theoretical results revealed acceleration to be a significant factor affecting the stalk-pulling process. Thus, acceleration was taken into account to increase the accuracy of the mechanical model and yield a more accurate description of the formation mechanism of the three states of stalk pulling. This optimized V-shaped toothed roller model was then applied in a field experiment. Specifically, an orthogonal experiment was carried out to optimize the rotational speed, cogging angle, and ground clearance of V-shaped toothed rollers with respect to the removal ratio. It is believed that the findings of this study can be used as a reference for the development of a highly efficient cotton stalk harvester.

## 2. Materials and Methods

### 2.1. V-Shaped Toothed Roller Stalk Pulling Structure and Working Principle

The V-shaped toothed roller stalk-pulling component, which mainly consists of a V-shaped toothed roller, stalk-clearing roller, reel wheel, and land wheel, as shown in Figure 1, is a key component of stalk-pulling machines. The key parameters are listed in Table 1. The V-shaped toothed plates are evenly distributed in three rows along the

circumferential direction of the V-shaped toothed roller and will hereafter be referred to as the toothed plates.

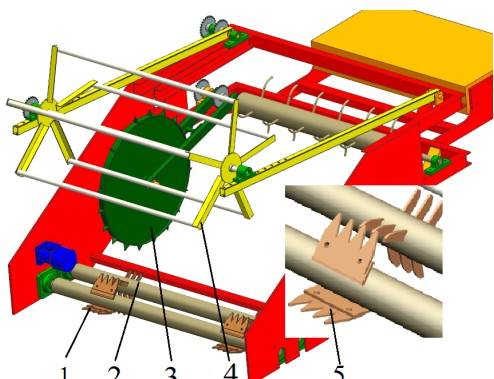

**Figure 1.** Structural schematic of V-shaped toothed roller. 1. V-shaped toothed roller, 2. Stalk-clearing roller, 3. Land wheel, 4. Reel wheel, 5. V-shaped toothed plate.

**Table 1.** The key parameters of the V-shaped toothed roller.

| Parameters | Value |
| --- | --- |
| Length × width × height (mm) | 1500 × 1200 × 1100 |
| Number of lines | 2 |
| Drive mode | hydraulic |
| Tooth plate: diameter × thick (mm) | 220 × 120 |
| Rotational speed (r/min) | 100–600 |

During the pulling process, the reel wheel rotates to feed the upper part of the cotton stalk into the harvester and the V-shaped toothed roller moves forward and rotates to embed the cotton stalks with lengths within the harvest width into the V-shaped toothed plate through the appropriate space between the V-shaped toothed rollers. The reel wheel works with the V-shaped toothed roller to apply push–pull forces that ultimately extract the cotton stalks embedded in the toothed plates from the soil.

### 2.2. Key Component Design

### 2.2.1. Determination of V-Shaped Toothed Roller Rotational Speed

The rotational speed of a V-shaped toothed roller should satisfy the following two conditions [10]: (1) the pulled-out, undetached cotton stalk must not hinder the operation of the adjacent toothed plate and (2) when the first toothed plate fails to pull out a cotton stalk, the second or third row of toothed plates must be able to embed and hold the cotton stalk (Figure 2). Considering that the driving speed of a tractor in a field ranges from 0.7 to 1.4 m/s and the plant spacing between cotton stalks typically ranges between approximately 18 and 25 cm, according to Equation (1), the rotational speed of the V-shaped toothed roller should exceed 156 r/min. The results of experimental tests indicate that the rotational speed of a V-shaped toothed roller should not be too high (i.e., it should be less than 500 r/min). This is because an excessively high rotational speed was found to be associated with excessive force being applied to the cotton stalk by the toothed plate, leading to a significantly higher stalk pull-out miss rate and fracture rate.

$$n_0 = \frac{1000 \times 60 \times v_0}{l_0} \times \frac{2}{3} = \frac{4000 v_0}{l_0} \tag{1}$$

$n_0$—rotational speed of tooth roller;
$v_0$—driving speed of tractor;
$l_0$—line spacing of cotton stalk.

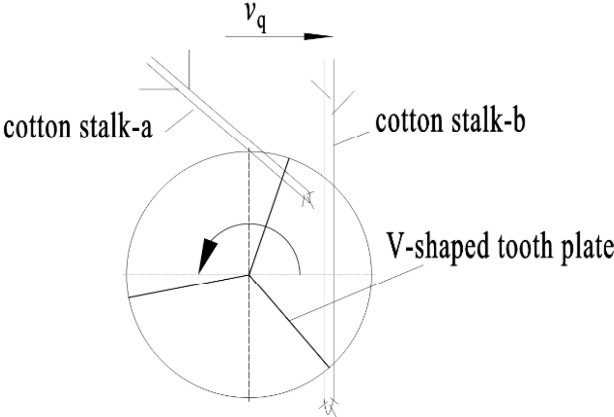

**Figure 2.** Schematic diagram showing the pull-out process of cotton stalk.

2.2.2. Determination of V-Shaped Toothed Plate Parameters

The maximum diameter of a cotton stalk is 22.75 mm; however, according to the results of multiple field measurements, the diameter of most cotton stalks in Wudi County, Shandong Province ranges from 10 to 18 mm. Thus, we mainly took cotton stalks with a diameter larger than 10 mm into consideration in the proposed V-shaped toothed roller stalk-pulling machine design. Note that the tooth width of the toothed plate was designed to be 30 mm. Additionally, the toothed roller was also designed to satisfy three requirements: (1) to improve the cotton stalk embedding and holding effectiveness, (2) to ensure that each cotton stalk remains embedded and held once it is extracted and swung backward, and (3) to ensure that the stalk-clearing roller can easily clear away each extracted cotton stalk from the toothed roller. The stress analysis results for cotton stalk extraction via the proposed V-shaped toothed roller are shown in Figure 3.

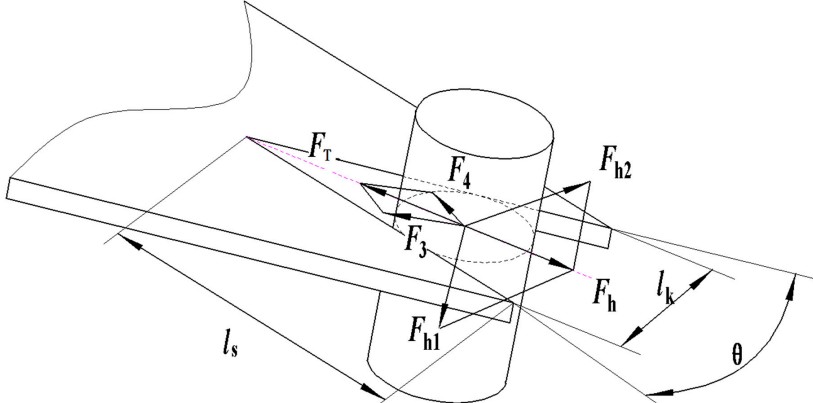

**Figure 3.** Stress analysis sketch of cotton stalk pulling.

According to the analysis of Figure 3, Formulas (2)–(4) can be obtained:

$$F_h = 2F_{h1} \cos(\frac{\pi}{2} - \frac{\theta}{2}) = 2F_{h1} \sin\frac{\theta}{2} \tag{2}$$

$$F_T = 2F_3 \cos\frac{\theta}{2} = 2F_{h1} \tan\alpha_0 \cos\frac{\theta}{2} \tag{3}$$

$$F_h = F_T \tag{4}$$

$\theta$—cogging angle (°);

$F_h$—combined force of pressure on cotton stalk produced by both sides of tooth cogging (N);

$F_3$—the friction of cotton stalk under pressure $F_{h1}$ (N);
$F_4$—the friction of cotton stalk under pressure $F_{h2}$ (N);
$F_T$—the combined force of friction (N);
$\alpha_0$—the frictional angle between cotton stalk and tooth plate (°).

The simultaneous Formulas (2)–(4) show that the above formula holds when $\theta = 2\alpha_0$. The frictional angle between the cotton stalk and the tooth plate can be converted by the friction coefficient, which could refer to the friction coefficient of 0.2~0.35 between wood and steel [24]. The converted $\alpha_0$ value was 11.3°~19.3°, thus the value of $\theta$ was 22.6°~38.6°.

The depth of cogging can be calculated from Formula (5):

$$l_s = \frac{l_k}{2\tan\frac{\theta}{2}} = \frac{15}{\tan\frac{\theta}{2}} \tag{5}$$

When the cogging has a smaller angle and a long and sharp tooth tip, the strength of the tooth plate is reduced. Therefore, the cogging angle was suggested to be greater than or equal to 25° after comprehensive consideration ($\theta = 25°$, $l_s = 67$ mm). To sum up, the range of the cogging angle is set at $25° \leq \theta \leq 40°$.

### 2.3. Cotton Stalk Pulling Process: Analysis of Toothed Roller Motion Trajectory

During operation, the V-shaped toothed roller, i.e., the main functioning component, makes forward and circular motions. As shown in Figure 4, the process of stalk pulling can be divided into the following four stages according to the motion trajectory: the clamping stage, pulling stage, delivering stage, and detaching stage. The analysis of tooth roller trajectory will provide a theoretical reference of the parameters design in the subsequent stalk removal mechanism. The motion trajectory formula for a certain point on the toothed plate is as follows:

$$\begin{cases} x = v_q t + \frac{R}{1000}\sin\left(\frac{\pi n t}{30}\right) \\ y = \frac{R+h}{1000} + \frac{R}{1000}\cos\left(\frac{\pi n t}{30}\right) \end{cases} \tag{6}$$

In the formula,
$v_q$—speed of forward motion (m/s);
$R$—radius of gyration of V-shaped tooth roller (mm);
$h$—height of central axis of V-shaped tooth bar above the ground (mm);
$n$—rotation speed of V-shaped tooth roller (rad/min);
$t$—time (s).

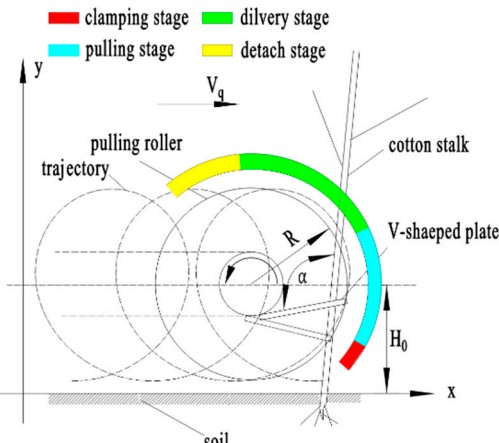

**Figure 4.** Motion trajectory sketch.

The simulation curves at different speeds and comparison chart of stalk pulling trajectory are exhibited in Figures 5 and 6. It can be clearly seen that the stalk-pulling trajectory is different at different speeds. Assuming that the cotton stalk is contacted and clamped at M, the machine advanced a distance of S and reaches N. It can be obtained from

Figure 6 that for the cotton stalk with different clamping heights, as the speed increases, the stalk pulling distance can be increased faster.

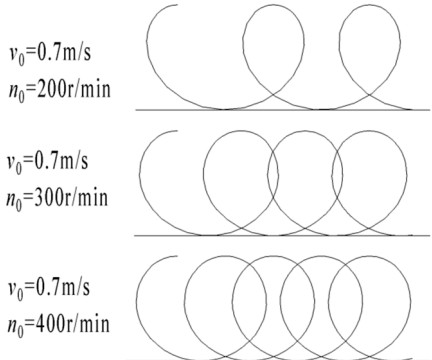

**Figure 5.** Simulation curves at different speeds.

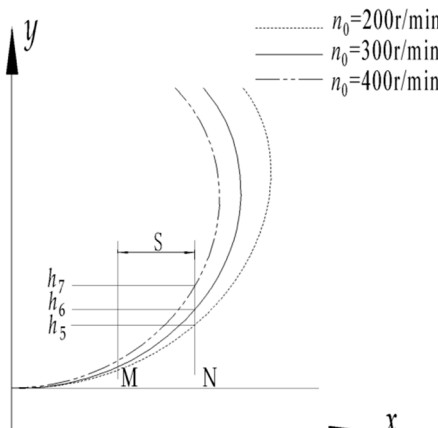

**Figure 6.** Comparison chart of stalk pulling trajectory.

### 2.3.1. Cotton Stalk Pulling Process: Collision Analysis

The mechanical changes that occur as the V-shaped toothed roller extracts stalks are very complex; these complex changes occur because of the interaction between the cotton stalk and the toothed plate and between the cotton stalk and soil. Thus, in this study, a simplified mechanical model was established by performing static analysis using data corresponding to a certain moment during the pulling process [21].

The type of collision between the rotating toothed plate and each cotton stalk was assumed to be elastic under ideal conditions [25], meaning that the mechanical properties of each cotton stalk that were temporarily altered as a result of deformation can be recovered without heating, sound, or kinetic energy loss. As such, the deformation of the cotton stalk was set to have a deformation stage and recovery stage. As shown in Figure 7, the force applied by the toothed plate was converted into deformation energy in the deformation stage; in the recovery stage, this deformation energy was defined as the bilateral forces applied to the cotton stalk by the toothed plate, i.e., $F_{h1}$ and $F_{h2}$. Under the action of $F_{h1}$ and $F_{h2}$, the cotton stalk and toothed plate moved relative to each other, generating frictional forces, $F_{1'}$ and $F_{2'}$, between them along the direction of the axis, as well as the corresponding resultant force, $F_{12}$.

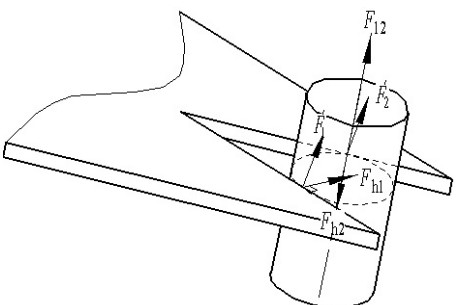

**Figure 7.** Sketch of the forces applied to a cotton stalk by the toothed plate.

2.3.2. Cotton Stalk Pulling Process: Mechanical Analysis of Cotton Stalk Push and Pull Forces

The toothed plate was designed to clamp around the cotton stalk after the initial collision to enable extraction by applying push and pull forces that can overcome the soil resistance. This stage included the following two processes: the clamping process and the push and pull process. The forces considered in the clamping process stress analysis for a single cotton stalk are illustrated in Figure 8. In general, the toothed plate initially makes contact with the phloem of the cotton stalk before extruding it. Then, the phloem applies bilateral forces, i.e., $F_{h1}$ and $F_{h2}$, on the toothed plate during the deformation recovery stage. As the machine moves forward, the toothed plate pushes the cotton stalk forward. During this time, the cotton stalk xylem begins undergoing flexible deformation and the push and pull process is initiated. The bilateral forces applied to the toothed roller by the xylem were set as $F_{h3}$ and $F_{h4}$.

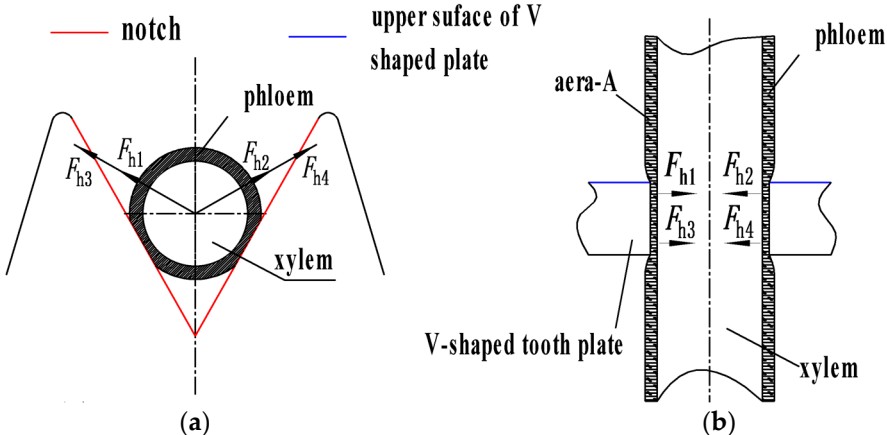

(**a**)  (**b**)

**Figure 8.** Sketches of cross section and longitudinal plane of cotton stalk clamped by the toothed plate: (**a**) schematic diagram of cross section under clamping state and (**b**) schematic diagram of longitudinal section in a clamped state.

During the push and pull process, the toothed plate continuously exerts force on the cotton stalk, consequently significantly deforming the phloem and creating the expanded zone A. At this moment, the cotton stalk is in an extrusion state, as shown in Figure 9. Thus, the pulling force applied to a single cotton stalk is the resultant force of (1) the upward force $F_s$ exerted on the cotton stalk by the toothed plate, which is perpendicular to the surface of the toothed plate; and (2) the bilateral frictional forces $F_{12}$ and $F_{34}$ applied to the cotton stalk by the clamping tooth, which are directed upward along the length of the cotton stalk.

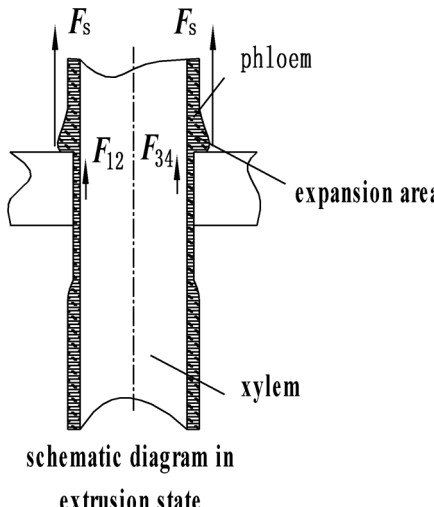

**Figure 9.** Sketch of longitudinal plane of cotton stalk extruded by the toothed plate.

2.3.3. Cotton Stalk Pulling Process: Analysis of the Bilateral Forces Acting on the Clamping Tooth

Dry friction can be defined as the force that acts to resist the relative motion of two solid objects. In the proposed system, there are two points at which friction occurs between the clamping teeth and the cotton stalk. One is the point at which the bilateral forces $F_{h1}$ and $F_{h2}$ are applied to the clamping tooth by the cotton stalk during the deformation recovery stage. The other is the point at which the cotton stalk undergoing flexible deformation applies bilateral forces to the clamping tooth because of being pushed and pulled. These bilateral forces have been defined as $F_{h3}$ and $F_{h4}$. The respective interactions between the cotton stalk and soil and the toothed plate and reel could abstract the cotton stalk into a simple beam, wherein the force-exerting points correspond to the contact points between the cotton stalk and the toothed plate. Thus, assuming that the contact points between the cotton stalk and soil, reel wheel and cotton stalk, and toothed plate and cotton stalk are A, B, and C, respectively, the force exerted by the toothed plate on the cotton stalk is P. A schematic showing the application of the simple beam theory to illustrate these forces and corresponding contact points is presented in Figure 10. It can be seen that within a certain range, the greater the impact intensity, the greater the deformation of the cotton stalk, and the greater the $F_{h1}$ and $F_{h2}$ produced by the elastic recovery force of the cotton stalk.

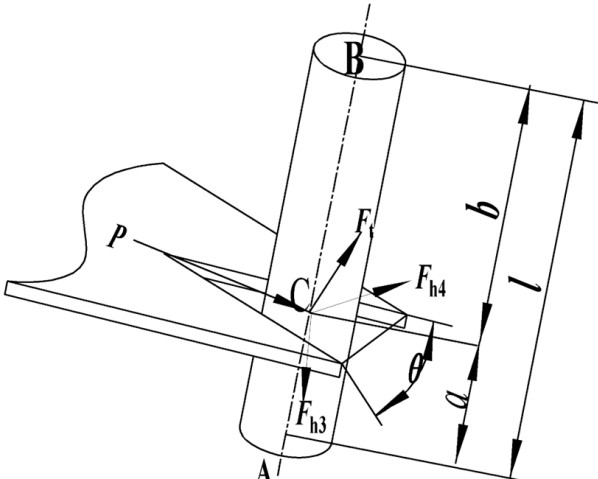

**Figure 10.** Schematic illustrating simple beam theory application.

According to the stress analysis in Figure 8 and the theory of simple beam, the flexible deformation $\Delta y$ of cotton stalk was calculated as follows:

$$\Delta y = \frac{Pab}{6lEJ}(a^2 + b^2 - l^2) \tag{7}$$

Through the transformation of the above formula, the force $P$ of the tooth plate exerted on cotton stalk was obtained as:

$$P = \frac{6\Delta ylEJ}{ab(a^2 + b^2 - l^2)} \tag{8}$$

In the formula,
$\Delta y$—the flexible deformation (mm);
$E$—elastic modulus (MPa);
$J$—moment of inertia (kg·m$^2$);
$l$—contact point height of reel wheel (mm);
$a$—ground clearance of the contact point between tooth plate and cotton stalk (mm);
$b$—space of contact points among reel wheel, tooth plate, and cotton stalk (mm);
$P$—the resultant force on the cotton stalk exerted by the tooth plate (N).

According to the parallelogram rule of force, the forces $F_{h3}$ and $F_{h4}$ of the resultant force $P$ on the sides of the tooth plate can be deduced as shown in Formula (9):

$$F_{h3} = F_{h4} = \frac{P \sin(\frac{\pi}{2} - \frac{\theta}{2}) \sin\theta}{\sin\theta} = \frac{p \cos\frac{\theta}{2}}{2 \sin\frac{\theta}{2} \cos\frac{\theta}{2}} = \frac{p}{2 \sin\frac{\theta}{2}} \tag{9}$$

Under the action of $P$, the maximum friction exerted by the tooth plate on the cotton stalk was $2F_{h3}$, which was along the axis of the cotton stalk. To sum up, the friction $F_t$ between the bilateral sides of the cogging and the cotton stalk was shown in Formula (10):

$$F_t = (F_{12} + F_{34})f = 2(F_{h1} + F_{h3})f = 2F_{h1}f + \frac{P}{\sin\frac{\theta}{2}}f \tag{10}$$

In the formula,
$F_t$—friction between the bilateral sides of the cogging and the cotton stalk, N;
$\theta$—angel of cogging, °;
$f$—static friction coefficient between the cotton stalk and tooth plate;
$F_{h3}$—force exerted by cotton stalk under deformation on bilateral sides of cogging, N;
$F_{h4}$—force exerted by cotton stalk under deformation on bilateral sides of cogging, N.

From Formulas (7)–(10), it can be concluded that under a certain value of $\Delta y$, the smaller the value of $a$ is, the larger the value of $P$ is. In the case of the same deformation of the cotton stalk, the lower the height of the clamped cotton stalk, the greater the force between the tooth plate and the cotton stalk, and the greater the friction that can be provided.

### 2.3.4. Cotton Stalk Pulling Process: Establishment of Mechanical Model

Generally, the forces exerted on the cotton stalk root by the soil act to resist the stalk-pulling process; this force system is quite complex. Therefore, for the purpose of simplification, the soil force system was modeled to have a resultant force, which is referred to as the cotton stalk soil resistance and denoted as $F_b$. $F_b$ was determined to occur along the length of the cotton stalk.

As the cotton stalk is being pulled, its initial state of static equilibrium is disrupted by sudden movement, which involves an acceleration component. This acceleration was set as $a_1$, the force producing the acceleration, which is equivalent to the resultant force $P$, was set as $F_a$, the frictional force was expressed as $F_t$, the extrusion force was set as $F_s$, and the soil resistance was set as $F_b$. Additionally, the angle between the force vector $F_a$

and the surface of the toothed plate was set as $\gamma$. A schematic illustrating the force system associated with this state of transition is presented in Figure 11.

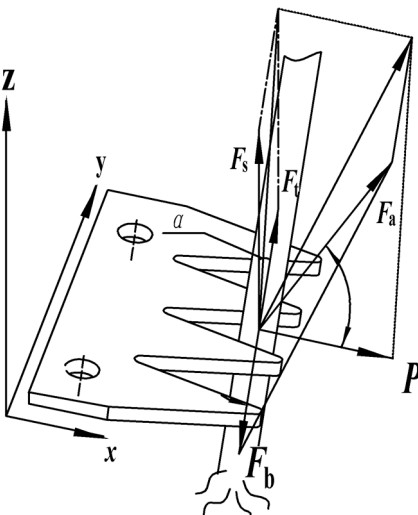

**Figure 11.** Schematic of forces acting on a cotton stalk transitioning from its initial static equilibrium to a dynamic state.

From the stress analysis of the cotton stalk in Figure 11, Formulas (11) and (12) can be obtained according to the dynamic knowledge such as D'Alembert's principle while taking the acceleration into consideration:

$$F_a \sin \gamma + F_b = F_s \cos \alpha + F_t \tag{11}$$

$$F_a = a_1 m = \frac{v_2 - v_1}{t_1} m = \frac{v_2}{t_1} m \tag{12}$$

By (10) and (11), (13) can be obtained:

$$m a_1 \sin \gamma + F_b = F_s \cos \alpha + 2F_{h1} + \frac{P}{\sin \frac{\theta}{2}} f \tag{13}$$

By (8), (12)–(14) can be obtained:

$$m \frac{v_2}{t_1} \sin \gamma + F_b = F_s \cos \alpha + 2F_{h1} f + \frac{6 \Delta y J e l f}{ab(a^2 + b^2 - l^2) \sin \frac{\theta}{2}} \tag{14}$$

In the formula,

$a_1$—the acceleration (m/s$^2$) (in actual conditions, the acceleration of cotton stalk a1 is a variable, which is related to the position and angle of the cotton stalk clamping. It is simplified for analysis here);

$m$—weight of cotton stalk (kg);

$v_2$—speed of the cotton stalk when being pulled out (m/s);

$v_1$—speed of cotton stalk at the static state (m/s, set as 0 m/s here);

$t_1$—time for the pull-out of the cotton stalk (s);

$F_a$—force-producing acceleration, equal to the combined force of the tooth plate to the cotton stalk and the soil to the cotton stalk (N).

Formula (14) was used to model and analyze the extraction, missed extraction, and fracture states of the cotton stalk pulling process. The following conclusions were made:

$F_{h1}$ generation is mainly dependent on the collision strength between the toothed plate and cotton stalk; more specifically, the elastic recovery of the cotton stalk is dependent on the speed of the collision. Within a certain range, a higher speed corresponds to larger

deformation of the cotton stalk. Additionally, the forces associated with deformation recovery are strong and serve to facilitate the cotton stalk clamping process. Upon making contact, when the effects of the collision between the toothed plate and cotton stalk cannot be endured by the cotton stalk, phloem rupture, a missed extraction, or even cotton stalk fracture can occur, as shown in Figure 12c. Under these conditions, the cotton stalk is broken, and the root remains in the soil. Thus, the rotational speed of the V-shaped roller should not be too low or too high.

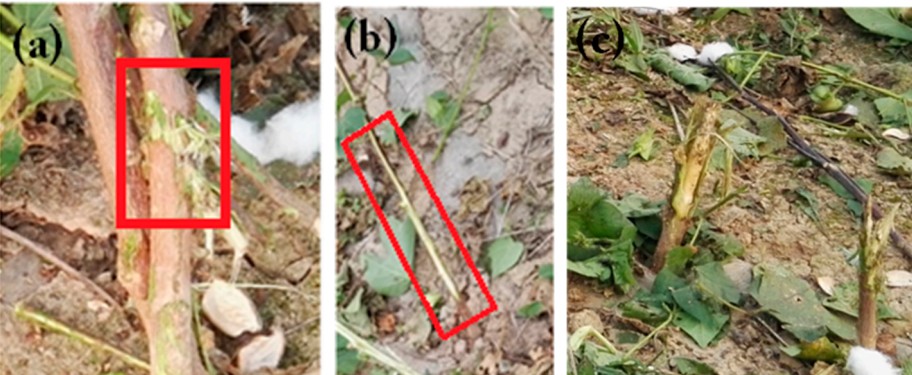

**Figure 12.** Conditions of the cotton stalk epidermis following a stalk extraction attempt. (**a**) Example of the epidermis following successful cotton stalk extraction. (**b**) Example of the epidermis of a cotton stalk that failed to be extracted. (**c**) Example of the epidermis of broken cotton stalks.

$F_s$ is the upward force of the toothed plate that acts against the cotton stalk. The effects of $F_s$ are mainly concentrated on the phloem of the cotton stalk. Thus, the magnitude of $F_s$ is primarily dependent on the characteristics of the cotton stalk phloem; this may be a significant reason for considerable differences in the stalk-pulling effects between autumn and spring.

Conclusions ① and ② above could be used to explain the phenomenon depicted in Figure 10. Figure 12a shows an example of the state of the cotton stalk epidermis at the V-shaped toothed plate clamping position after a cotton stalk was successfully extracted in autumn. The epidermis was obviously ruptured; however, in terms of length, the rupture was not remarkably substantial. (Most ruptures were less than 25 mm in length according to the results of rough statistics.) Figure 12b shows the state of the epidermis of a cotton stalk that was unsuccessfully extracted. It can be seen that a large area of the epidermis was destroyed, and some side branches were broken. This state is referred to as the missed extraction state.

In Formula (14), *a* represents the ground clearance of the contact point between the toothed plate and cotton stalk. Theoretically, a lower value of *a* should correspond to a better cotton stalk-pulling outcome. $\theta$ is the angle between the toothed plates. When *P* is constant, a lower value of $\theta$ corresponds to better gripping force.

Following the analysis of these results, successful cotton stalk extraction has been determined to be associated with a relative displacement between the V-shaped toothed plate and cotton stalk that does not exceed 25 mm. This was determined based on whether the extrusion force $F_s$ exceeded the endurable limit of the cotton stalk. Thus, the extrusion force $F_s$ should be maintained at an appropriate value during the stalk-pulling process. Furthermore, according to Formula (13), under the condition that the soil resistance $F_b$ is constant, the cotton stalk pulling outcome can be improved by reducing the speed $v_2$, the stalk-pulling height a, or the cogging angle $\theta$.

$$m\frac{v_2\downarrow}{t_1}\sin\gamma + F_b = F_s\cos\alpha\downarrow + 2F_{h1}f\uparrow + \frac{6\Delta y Jelf}{ab(a^2\downarrow + b^2 - l^2)\sin\frac{\theta\downarrow}{2}}\uparrow \tag{15}$$

The left side of Formula (15) describes the forces of the soil acting on the cotton stalk root, as well as the force associated with cotton stalk acceleration, which has been set as a passive load. The right side of the equation describes the force exerted by the toothed plate on the cotton stalk; it has been set as the active force. Among all the parameters influencing the active force, the bilateral forces $F_{h1}$ and $F_{h2}$ applied to the clamping tooth by the cotton stalk during the deformation recovery stage of the phloem, the ground clearance $\alpha$, and the cogging angle $\theta$ are controllable. With the exception of the speed of the toothed roller $n$ that influences $F_{h1}$ and $F_{h2}$ that is adjustable, the remaining parameters, such as the friction coefficient $f$, elastic modulus $E$, cotton stalk height $l$, and extrusion force $F_s$ cannot be controlled. Thus, in this study, the speed of the toothed roller $n$, ground clearance $\alpha$, and cogging angle $\theta$ were determined to be test factors.

*2.4. Field Tests*

2.4.1. Test Equipment and Materials

To determine the working parameters for toothed rollers, a V-shaped toothed roller stalk-pulling device bench was developed, and field tests were conducted (Figure 13). The test site was the Demonstration Base of New Cotton Variety K836 for Simplified and High-Yield Cultivation in Binzhou City, Shandong Province. The test was conducted in spring. The cotton variety is China Cotton Institute—50 (CCRI50). The cotton stalk rows were spaced at 76 cm, and the target plant spacing was 20 cm, with an average of 30 cm. The height of the cotton stalks typically ranged between 95 and 105 cm, and the diameter of the cotton stalks ranged from 10 to 22 cm. The average soil firmness is 672.3 KPa and the average soil moisture content is 27.4%.

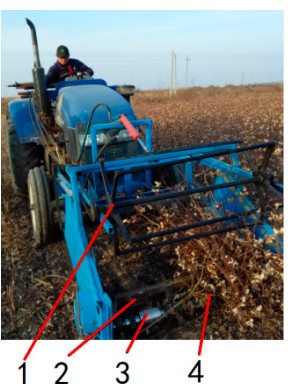

**Figure 13.** Field test photo: 1. Reel wheel 2. V-shaped tooth roller 3. Torque transducer 4. Cotton stalks.

2.4.2. Test Factors and Levels

The test factors were the speed $n$ of the toothed roller, ground clearance $\alpha$, and cogging angle $\theta$. According to the results of calculations, the speed of the V-shaped toothed roller varied between 176 and 500 r/min; thus, the test speeds were preliminarily determined to be 200, 300, and 400 r/min. The results of the theoretical analysis indicated that operating the toothed roller at a higher ground clearance yields better performance; thus, $\alpha$ was assigned values of −20 mm (i.e., 20 mm into the soil), 20 mm, and 60 mm in this study. Based on the optimized above-described structural design, the cogging angle $\theta$ should range between 25° and 38.6°. However, to account for the coefficient of friction between the cotton stalk and steel plate and to minimize processing difficulty, the maximum cogging angle was set to 40° for the field test. In the case of a certain thrust, the smaller the cogging angle, the greater the clamping force on the cotton stalk. However, there are two uncertainties: 1. Too small an angle leads to too long tooth height, which easily causes the insufficient strength of tooth plate structure; 2. Excessive clamping force will easily cause the cotton stalks to fall off easily. Thus, the cogging angles implemented in the field test were 25°, 32.5°, and 40°. The test levels are summarized in Table 2. The $L_9\ (3^4)$ orthogonal

experiment was designed to have nine test groups. The experiment was repeated three times for each group, and the mean value was calculated.

**Table 2.** Factors and levels of orthogonal experiment.

| Levels | Factors | | |
|---|---|---|---|
| | A. Ground Clearance/(mm) | B. Rotation Speed/(r/min) | C. Cogging Angle/(°) |
| 1 | −20 | 200 | 25 |
| 2 | 20 | 300 | 32.5 |
| 3 | 60 | 400 | 40 |

### 2.4.3. Test Methods

The test was carried out in accordance with the standard GB/T8097-2008 [26]. To ensure stable operation of the equipment and minimize error, the puller machine was allowed to adjust its operating posture by traveling a distance of 20 m before data were collected. To facilitate the process of data collection, two rows of cotton stalks were harvested each time. The data collection region was 30 m long, corresponding to approximately 200 cotton stalks. The evaluation index for the test was the removal rate; it was determined as follows:

$$y = \frac{M_b}{M_z} \times 100\% \tag{16}$$

In the formula, $y$—the removal rate of cotton stalks, %;
$M_b$—the number of cotton stalks pulled out, plant;
$M_z$—the sum of cotton stalks, plants.

## 3. Results and Discussion

The test plan and results were shown in Table 3 (A, B, and C are the ground clearance, the rotation speed, and the cogging angle, respectively).

**Table 3.** Test results and analysis.

| Experiment No. | | Factors | | | | $y$ |
|---|---|---|---|---|---|---|
| | | A | B | C | D (Emptyrow) | Removal Rate/% |
| 1 | | 1 | 1 | 1 | 1 | 94.76 |
| 2 | | 1 | 2 | 2 | 2 | 97.34 |
| 3 | | 1 | 3 | 3 | 3 | 93.29 |
| 4 | | 2 | 1 | 2 | 3 | 89.58 |
| 5 | | 2 | 2 | 3 | 1 | 92.77 |
| 6 | | 2 | 3 | 1 | 2 | 90.9 |
| 7 | | 3 | 1 | 3 | 2 | 82.53 |
| 8 | | 3 | 2 | 1 | 3 | 85.71 |
| 9 | | 3 | 3 | 2 | 1 | 80.93 |
| Average removal rate/% | $k_1$ | 95.13 | 88.96 | 90.46 | 89.49 | |
| | $k_2$ | 91.08 | 91.94 | 89.28 | 90.26 | |
| | $k_3$ | 83.06 | 88.37 | 89.00 | 89.53 | |
| | $R$ | 12.07 | 3.57 | 1.46 | 0.77 | |
| Influence order: $A > B > C$ | | | | | | |

According to the results of numerical analysis for the range $R$ for each factor in Table 2, it can be seen that the order of significance in terms of the influence of each factor on the evaluation index (i.e., the removal rate) was $A > B > C$. The ground clearance most significantly influenced the removal rate, followed by the rotational speed, and the cogging angle. To optimize the combination of these factors to obtain the highest cotton stalk removal rate, the removal rate was plotted as a function of the averaged test factor results presented in Table 2; the resulting graph is shown in Figure 14. It can be ascertained from

Figure [12] that the combination of factors that yielded the highest removal rate was $A_1B_2C_1$, i.e., the combination of a ground clearance of $-20$ mm, a rotational speed of 300 r/min, and cogging angle of 25°.

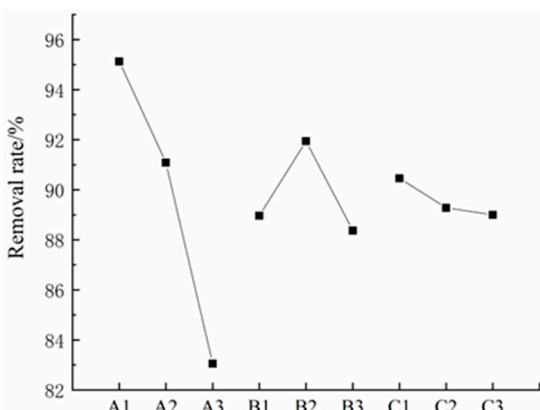

**Figure 14.** Relationship between factor level and removal rate.

Minitab software was used as a platform to analyze the variance of the results of the orthogonal experiment and quantify the respective influences of the three factors (i.e., cogging angle, rotational speed, and ground clearance) on the removal rate; the results are summarized in Table [4].

**Table 4.** Analysis of variance for orthogonal experiment results.

| Source | Sum of Squares | Degree of Freedom | Sum of Mean Squares | F Value | Significant Level $p$ |
|---|---|---|---|---|---|
| A | 226.568 | 2 | 113.284 | 200.97 | 0.005 ** |
| B | 21.962 | 2 | 10.981 | 19.48 | 0.049 * |
| C | 2.296 | 2 | 1.148 | 2.04 | 0.329 |
| Pure error | 1.127 | 2 | 0.564 | | |
| Cor total | 251.954 | 8 | | | |

Note: $p < 0.01$ (highly significant, **); $p < 0.05$ (significant, *). Based on the analysis of variance results presented in Table [3], it can be ascertained on the basis of FA > FB > FC that A had an extremely significant influence on the removal rate, B had a significant influence, and C had minimal influence. Such results are consistent with the range analysis results. In addition, it can be also ascertained from Table [3] that the sum of squares error was much less than that of factor A, factor B, or factor C, indicating that the correlations between test factors did not considerably affect the evaluation index.

Compared with the knife-roller type stalk pulling operation equipment, it has the advantages of not entering the soil and reducing energy consumption on the basis of non-alignment operation. Tests have proved that in the Binzhou area, cotton stalks with larger diameters can be effectively pulled out. However, since this type of stalk-pulling device has many V-shaped structures on its stalk-pulling rollers, once cotton stalks are entangled, the cotton stalks will be damaged. The damaged cotton stalk cannot fall off, which leads to its working failure; secondly, this device has been tested in a densely planted cotton area. When the average root diameter of cotton stalks is about 10 mm, the stalk-pulling effect is very poor, so this device still needs further in-depth research. The most foreign equipment introduced in the literature is the double-roller type stalk pulling equipment. Our team has used the domestically produced double-roller equipment to conduct experiments in the Binzhou area, and the effect is not ideal. The variety and planting mode of cotton stalks have a great influence on the mechanical properties of cotton stalks. Up to now, there is no stalk-pulling equipment that can adapt to cotton stalks in different regions. Therefore, further research is needed to solve the issue of stalk-pulling technology.

## 4. Conclusions

1.  Combining parameters such as cotton diameter, plant distance, V-shaped tooth speed, and friction coefficient between cotton and tooth plate, it is determined that the suitable angle range of the groove was 25.6° to 38.6°.

2.  On the basis of the cotton stalk pulling model, acceleration was further introduced. A detailed theoretical explanation was given for the state of missing or breaking stalks during pulling.

3.  The orthogonal test results demonstrated that the ground clearance of the V-shaped toothed roller had an extremely significant effect on the removal rate. The lower the ground clearance, the better the soil penetration effect. The tooth groove angle has no significant effect on the extraction rate. The effect test showed that the primary and secondary order of the influence of various factors on the cleaning rate was ground clearance > speed > cogging angle. Additionally, under the condition of not considering the extraction energy consumption, the more suitable combination of mechanism parameters was ground clearance −20 mm. The speed of the V-shaped toothed roller was 300 r/min, and the angle of the tooth groove was optional between 25° and 40°.

**Author Contributions:** Conceptualization, Z.W. and W.Z.; methodology, Z.W., W.Z. and J.F.; software, Z.W. and W.Z.; validation, Z.W., W.Z., J.F. and H.X.; formal analysis, Z.W. and W.Z.; investigation, Z.W., W.Z., J.F. and H.X.; data curation, Z.W., W.Z. and J.F.; writing—original draft preparation, Z.W., W.Z. and J.F.; writing—review and editing, J.F, Y.Z. and M.C.; visualization, Z.W., Y.Z. and M.C.; supervision, Y.Z. and M.C.; funding acquisition, M.C. All authors have read and agreed to the published version of the manuscript.

**Funding:** This work was financially supported by the National Natural Science Foundation of China (Grant No. 51505242) and the Innovative Research Group of Agricultural Production Waste Resource Utilization Equipment.

**Institutional Review Board Statement:** Not applicable.

**Data Availability Statement:** Data are contained within the article.

**Acknowledgments:** The authors thank the editor and anonymous reviewers for providing helpful suggestions for improving the quality of this manuscript.

**Conflicts of Interest:** The authors declare no conflict of interest.

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
