# Peer review of "V-Shaped Toothed Roller Cotton Stalk Puller: Numerical Modeling and Field-Test Validation"

_agriculture, doi:10.3390/agriculture13061157_

Round 1

Reviewer 1 Report

In this paper, a simplified mathematical model of V-shaped toothed roller sucker rod extractor is established, and mechanical analysis is carried out on the basis of this model to explore the causes of sucker rod error and fracture. It has certain practicability and research significance, but there are still some problems. Some problems are as follows.

1.”The results of theoretical research have shown that, although it is still in an exploratory phase, the V-shaped toothed roller type not only has the advantage of high removal ratio commonly associated with the knife-roller type but also the advantage of low energy consumption.”What are these bases, if there is a basis to indicate the relevant references.

2.The fonts used in the picture are not unified, and the tables need to be adjusted, such as Figs.2,4,11 ; table 1, 2, etc.

3.Why does the ground clearance and the cogging angle only choose three values ? What is the basis ? Additional explanation is suggested.

4. There are some irregularities in the writing of the article, such as formula 1, formula 12. Formula 14 has a comma, and the annotation of formula 15 for formula characters is not unified with other formulas.

5. The average removal rate obtained by further optimization of cotton straw is higher than that obtained by theoretical analysis. What are the factors that affect the results ?

6.The format at the end of the reference is incorrect, and some English words in the article are incorrect, hoping to correct.

The format at the end of the reference is incorrect, and some English words in the article are incorrect, hoping to correct.

Author Response

Response to Reviewer 1 Comments

In this paper, a simplified mathematical model of V-shaped toothed roller sucker rod extractor is established, and mechanical analysis is carried out on the basis of this model to explore the causes of sucker rod error and fracture. It has certain practicability and research significance, but there are still some problems. Some problems are as follows.

1.”The results of theoretical research have shown that, although it is still in an exploratory phase, the V-shaped toothed roller type not only has the advantage of high removal ratio commonly associated with the knife-roller type but also the advantage of low energy consumption.”What are these bases, if there is a basis to indicate the relevant references.

  • Response: The reference has been cited.
  1. Guo, Z.; Shi, J.; Kang, X. The Resistance Analysis of Roller-cotton Stalks. Journal of Agricultural Mechanization Research 2009, 31, 37-39.

2.The fonts used in the picture are not unified, and the tables need to be adjusted, such as Figs.2,4,11 ; table 1, 2, etc.

  • Response: All the figures and tables have been revised.

3.Why does the ground clearance and the cogging angle only choose three values ? What is the basis ? Additional explanation is suggested.

  • Response: The selection of the levels of the ground clearance was based on the theoretical analysis. And for the cogging angle, in the case of a certain thrust, the smaller the cogging angle, the greater the clamping force on the cotton stalk. But there are two uncertainties: 1. Too small an angle leads to too long tooth height, which is easy to cause the insufficient strength of tooth plate structure; 2. Excessive clamping force will easily cause the cotton stalks to fall off easily. The explanation has been added in the 2.4.2.

  1. There are some irregularities in the writing of the article, such as formula 1, formula 12. Formula 14 has a comma, and the annotation of formula 15 for formula characters is not unified with other formulas.
  • Response: Thanks for the good advice. All the commas have been deleted. And all the formats of formulas have been checked and revised.

  1. The average removal rate obtained by further optimization of cotton straw is higher than that obtained by theoretical analysis. What are the factors that affect the results ?
  • Response: The conclusion of this part is our carelessness. The optimized removal rate in this paper is higher than the research results of existing literature, not the theoretical analysis. The relevant discussion part has been rewritten and merged with the results part.

6.The format at the end of the reference is incorrect, and some English words in the article are incorrect, hoping to correct.

  • Response: Thanks for the good advice. The format and English words in the whole text have been carefully checked and revised.

Reviewer 2 Report

Asbstract should be improved to show the important of thsi study.

Minor editing of English language required

Author Response

Response to Reviewer 2 Comments

  1. In chapter "2.2.1 Determination of V-shaped toothed roller rotational speed", the minimum speed of V-shaped toothed roller is 176r/min, but the conditions given in the text do not directly lead to the above conclusion, so it is suggested to add relevant explanation or calculation process.
  • Response: The rotational speed was calculated according to the follow equation.

                        (1)

n0 - rotaional speed of tooth roller;

v0 - driving speed of tractor;

l0 - line spacing of cotton stalk.

  1. Does the forward speed of the machine have any influence on the force of the cotton stalks in chapter "2.2.2 Determination of V-shaped toothed plate parameters"?
  • Response: In the pulling process, the forward speed of the machine has influence on the force of the cotton stalks, the detailed explanation was presented in 2.3.1. The force analysis of cotton stalks related to the structural parameters of the tooth disc is explained in 2.2.2, without considering the state of motion.

  1. Chapter "2.3 Cotton-stalk pulling process: analysis of toothed roller motion trajectory" should indicate the purpose of the analysis of tooth roller trajectory. This demonstrates the significance and value of the study.
  • Response: The theoretical analysis of tooth roller trajectory is to understand the influence of the clamping height and the rotation speed of the tooth disc on the cotton stalk removal process, so as to provide a theoretical reference for the selection of the above parameter ranges in the subsequent stalk removal mechanism. And the purpose has been added in the first paragraph of 2.3.

  1. Chapter "2.3.1~2.3.3" has made the mechanical analysis of different parts respectively, and the relevant conclusions should be supplemented with the force changes under different stages.
  • Response: Thanks for the good advice. The relevant conclusions have been added. Please see the details in the manuscript.

  1. Is it necessary to consider the influence of soil type and soil moisture content in the field test to derive the optimal combination of parameters using the orthogonal test method? If so, please specify the soil conditions.
  • Response: The experiment in this paper was carried out in the cotton area of Binzhou, Shandong, and the soil types are not much different. Soil moisture content has a certain influence on the stalk pulling test. The main purpose of this paper is to study the influence of mechanism design and movement parameters on the stalk pulling process. In the follow-up, the influence of the operating environment on the stalk pulling will be studied. Information on cotton varieties and soil parameters for field trials in this article has been added.

Reviewer 3 Report

1. In chapter "2.2.1 Determination of V-shaped toothed roller rotational speed", the minimum speed of V-shaped toothed roller is 176r/min, but the conditions given in the text do not directly lead to the above conclusion, so it is suggested to add relevant explanation or calculation process.

2. Does the forward speed of the machine have any influence on the force of the cotton stalks in chapter "2.2.2 Determination of V-shaped toothed plate parameters"?

1.    Chapter "2.3 Cotton-stalk pulling process: analysis of toothed roller motion trajectory" should indicate the purpose of the analysis of tooth roller trajectory. This demonstrates the significance and value of the study.

4. Chapter "2.3.1~2.3.3" has made the mechanical analysis of different parts respectively, and the relevant conclusions should be supplemented with the force changes under different stages.

5. Is it necessary to consider the influence of soil type and soil moisture content in the field test to derive the optimal combination of parameters using the orthogonal test method? If so, please specify the soil conditions.

no

Author Response

Response to Reviewer 3 Comments

  1. In chapter "2.2.1 Determination of V-shaped toothed roller rotational speed", the minimum speed of V-shaped toothed roller is 176r/min, but the conditions given in the text do not directly lead to the above conclusion, so it is suggested to add relevant explanation or calculation process.
  • Response: The rotational speed was calculated according to the follow equation.

                        (1)

n0 - rotaional speed of tooth roller;

v0 - driving speed of tractor;

l0 - line spacing of cotton stalk.

  1. Does the forward speed of the machine have any influence on the force of the cotton stalks in chapter "2.2.2 Determination of V-shaped toothed plate parameters"?
  • Response: In the pulling process, the forward speed of the machine has influence on the force of the cotton stalks, the detailed explanation was presented in 2.3.1. The force analysis of cotton stalks related to the structural parameters of the tooth disc is explained in 2.2.2, without considering the state of motion.

  1. Chapter "2.3 Cotton-stalk pulling process: analysis of toothed roller motion trajectory" should indicate the purpose of the analysis of tooth roller trajectory. This demonstrates the significance and value of the study.
  • Response: The theoretical analysis of tooth roller trajectory is to understand the influence of the clamping height and the rotation speed of the tooth disc on the cotton stalk removal process, so as to provide a theoretical reference for the selection of the above parameter ranges in the subsequent stalk removal mechanism. And the purpose has been added in the first paragraph of 2.3.

  1. Chapter "2.3.1~2.3.3" has made the mechanical analysis of different parts respectively, and the relevant conclusions should be supplemented with the force changes under different stages.
  • Response: Thanks for the good advice. The relevant conclusions have been added. Please see the details in the manuscript.

  1. Is it necessary to consider the influence of soil type and soil moisture content in the field test to derive the optimal combination of parameters using the orthogonal test method? If so, please specify the soil conditions.
  • Response: The experiment in this paper was carried out in the cotton area of Binzhou, Shandong, and the soil types are not much different. Soil moisture content has a certain influence on the stalk pulling test. The main purpose of this paper is to study the influence of mechanism design and movement parameters on the stalk pulling process. In the follow-up, the influence of the operating environment on the stalk pulling will be studied. Information on cotton varieties and soil parameters for field trials in this article has been added.

Reviewer 4 Report

1. In 2018, the planting area of cotton reached approximately 50.285 million mu in China [1, 2];”: Do you have the data in recent years?      Also, mu?

2. “2.3. Cotton-stalk pulling process: analysis of toothed roller motion trajectory”: how to guide the design and optimization of key parts of this machine is not clear, please improve in the manuscript.

Also, please add the motion analysis and simulation for the pulling operation of this cotton-stalk puller, and it will be useful for the design and optimization of key parts.

Please add the photos to show the key parts of this machine.

3. ”4.Discussion” is too short to be a section. Please improve the discussion based on the structural features of the machine, experimental results and current studies in the world.

4. “Furthermore, a ground clearance of -20 mm, rotational speed of 300 r/min, and cogging angle of 25° yielded an average removal ratio of 98.27%”: Will the varieties of cotton affect this result?

Moderate editing of English language.

Author Response

Response to Reviewer 4 Comments

  1. “In 2018, the planting area of cotton reached approximately 50.285 million mu in China [1, 2];”: Do you have the data in recent years? Also, mu?
  • Response: The data of 2020 has been replaced. And the unit has been changed to “hectares”. Please see the detail in the manuscript.

  1. “2.3. Cotton-stalk pulling process: analysis of toothed roller motion trajectory”: how to guide the design and optimization of key parts of this machine is not clear, please improve in the manuscript.
  • Response: The theoretical analysis of tooth roller trajectory is to understand the influence of the clamping height and the rotation speed of the tooth disc on the cotton stalk removal process. The results of theoretical analysis showed that reducing the height of stalk pulling can improve the efficiency of stalk pulling, and changing the tooth shape can also change the clamping force of cotton stalks. The analysis of tooth roller trajectory will provide a theoretical reference of the parameters design in the subsequent stalk removal mechanism. Please see the detail in the manuscript.

Also, please add the motion analysis and simulation for the pulling operation of this cotton-stalk puller, and it will be useful for the design and optimization of key parts.

  • Response: The motion analysis and simulation of stalk pulling trajectory have been added. Please see details in Figure 5 and Figure 6.

Please add the photos to show the key parts of this machine.

  • Response: Number 5 in Figure 1 is an enlarged three-dimensional view of the V-shaped tooth plate, and the relevant parameter dimensions of the mechine are listed in Table 1.

  1. ”4.Discussion” is too short to be a section. Please improve the discussion based on the structural features of the machine, experimental results and current studies in the world.
  • Response: The discussion part has been combined with the results part, and the discussion is carried out in combination with the characteristics of the machine, the experimental results and the international status quo. Please see details in manuscript.

  1. “Furthermore, a ground clearance of -20 mm, rotational speed of 300 r/min, and cogging angle of 25° yielded an average removal ratio of 98.27%”: Will the varieties of cotton affect this result?
  • Response: The varieties of cotton would affect this result. In fact, the diameter of cotton stalks has a great influence on the removal ratio, and the diameters of different varieties of cotton stalks vary greatly. The device designed in this paper is mainly designed for the cotton variety Zhongmian Institute-50 (CCRI50) in Shandong.

Reviewer 5 Report

The problem studied in this paper is relevant and interesting. Overall, the manuscript is well organized and its presentation is good. However, some major issues still need to be improved:

(1) The novelty and motivation of the method should be emphasized clearly compared with previous related works.

(2) The abstract, conclusion and future scope should be clearly understood to the audience. Author must revise in the context of novelty and important findings of the proposed research work.

(3) Author should present some simulation results and indicate which type of tool/platform is using by the authors.

(4) “Specifically, the key components of the machine were optimized in an orthogonal experiment with the rotational speed, cogging angle, and ground clearance as the influencing factors, and the removal ratio as the evaluation index.” Which key components have been optimized?

(5) Author must compare with the existing techniques.

(6) The English and typo errors of the paper should be checked in the presence of native English speaker.

The English and typo errors of the paper should be checked in the presence of native English speaker.

Author Response

Response to Reviewer 5 Comments

The problem studied in this paper is relevant and interesting. Overall, the manuscript is well organized and its presentation is good. However, some major issues still need to be improved:

(1) The novelty and motivation of the method should be emphasized clearly compared with previous related works.

  • Response: Thanks for the good advice. The novelty of this work is that the designed machine is equipped with torque, speed sensors and other components, which can control the motion parameters very accurately, so as to facilitate the research of the experiment. Please see details in the manuscript.

(2) The abstract, conclusion and future scope should be clearly understood to the audience. Author must revise in the context of novelty and important findings of the proposed research work.

  • Response: The abstract has been revised, and the novelty has been revised in the last paragraph of “Introduction”.

(3) Author should present some simulation results and indicate which type of tool/platform is using by the authors.

  • Response: The simulation results have been added. Please see the Figure 5 and Figure 6.

(4) “Specifically, the key components of the machine were optimized in an orthogonal experiment with the rotational speed, cogging angle, and ground clearance as the influencing factors, and the removal ratio as the evaluation index.” Which key components have been optimized?

  • Response: This paper mainly studies parameters of V-shaped tooth plates. And this has been reviesed in the manuscript.

 (5) Author must compare with the existing techniques.

  • Response: The comparison with existing techniques has been discussed and added in the last paragraph of part 3. “Results and discussion”.

(6) The English and typo errors of the paper should be checked in the presence of native English speaker.

  • Response: The format and English words in the whole text have been carefully checked and revised.

Reviewer 6 Report

This thesis is aimed at the design of the V-shaped toothed cotton animal remover. Taking the speed, cogging angle, and ground clearance as the influencing factors, and the removal rate as the price index, the orthogonal test optimization of the key components of the whole machine was carried out. I suggest the following revisions to this manuscript:

1. The coordinate axes should be supplemented in Figure 3.

2. Should the 0 in the following sentence of reference [22] be a subscript?

3. Figure 5 and Figure 3 must be placed in the same position to compare and analyze the stress process.

4. Which parameters can be obtained from formulas 8 and 9? What do these parameters mean for the simulation?

no

Author Response

Response to Reviewer 6 Comments

This thesis is aimed at the design of the V-shaped toothed cotton animal remover. Taking the speed, cogging angle, and ground clearance as the influencing factors, and the removal rate as the price index, the orthogonal test optimization of the key components of the whole machine was carried out. I suggest the following revisions to this manuscript:

1. The coordinate axes should be supplemented in Figure 3.

  • Response: Figure 3 has been reviesed.

  1. Should the 0 in the following sentence of reference [22] be a subscript?
  • Response: α0 has been revised.

  1. Figure 5 and Figure 3 must be placed in the same position to compare and analyze the stress process.
  • Response: Those two figures have been placed in the same position.

  1. Which parameters can be obtained from formulas 8 and 9? What do these parameters mean for the simulation?
  • Response: Formulas 8 and 9 now was formulas 9 and 10. They are the derivation process. Formula 9 can show the influence of the angle of the V-shaped tooth plate on the clamping force. Formula 10 showed the friction between the V-shaped tooth plate and the cotton stalk when it lifts the cotton stalk.

Reviewer 7 Report

·         The paper is generally prepared in a standard format with sound literary and technical presentation. It is well written in clear and concise manner. The figures and schemes are properly presented. The language is comprehensive and coherent while errors and inaccuracies are relatively rare.

·         The paper is on the modeling and field validation of a V-shaped toothed roller cotton-stalk puller

However, the authors should make the following corrections to make the paper acceptable for further consideration:

·         Some cited papers are not listed. For example,  He et al. and Dai et al. on page 2

·         The paragraph before Section 2 (Materials and Methods) on Page 2  starting with “In thios study, to elucidate…” should be taken to section 2.

·         Section 4 (Discussion) is too shallow. The authors should use this section to discuss the results of modelling and field validation rather than briefly explaining how these were done.

·                    Section 5 (Conclusion) is too lengthy. it should end with the summary of your thoughts and convey the larger implications of your study. It shouldn’t be used to catalogue your methodology, results or discussion.

Author Response

Response to Reviewer 7 Comments

The paper is generally prepared in a standard format with sound literary and technical presentation. It is well written in clear and concise manner. The figures and schemes are properly presented. The language is comprehensive and coherent while errors and inaccuracies are relatively rare.

The paper is on the modeling and field validation of a V-shaped toothed roller cotton-stalk puller.

However, the authors should make the following corrections to make the paper acceptable for further consideration:

Some cited papers are not listed. For example,  He et al. and Dai et al. on page 2

  • Response: Those literatures have been added.
  1. He, X. Design and Experiment Research of Cotton Stalk Pulling-out Mechanism. Master; Hunan Agricultural University, 2016.
  2. Dai, Z. Design and Research on Cotton Stalk Drawing Institutions of Pull-out Cotton Stalk Mehanism. Master; Hunan Agricultural University, 2015.

The paragraph before Section 2 (Materials and Methods) on Page 2  starting with “In thios study, to elucidate…” should be taken to section 2.

  • Response: This paragraph has been revised, as to show the novelty and main content of this research.

Section 4 (Discussion) is too shallow. The authors should use this section to discuss the results of modelling and field validation rather than briefly explaining how these were done.

  • Response: Section 4 has been combined with section 3, and some comparisons and discussions have been added. Please see the last paragraph in section 3.

Section 5 (Conclusion) is too lengthy. it should end with the summary of your thoughts and convey the larger implications of your study. It shouldn’t be used to catalogue your methodology, results or discussion.

  • Response: The conclusion has been revised. Please see detail in the manuscript.

Round 2

Reviewer 4 Report

This paper has been improved according to the comments. The comments for the improvement of the manuscript are as follows:
1.Suggest modifying the color of the lines marked in Figures 12 and 13 to make them more prominent.

Author Response

Response to Reviewer 4 Comments

This paper has been improved according to the comments. The comments for the improvement of the manuscript are as follows:

1.Suggest modifying the color of the lines marked in Figures 12 and 13 to make them more prominent.

  • Response: Figures 12 and 13 have been revised.

Figure 12. Conditions of the cotton stalk epidermis following a stalk extraction attempt. (a) Example of the epidermis following successful cotton stalk extraction. (b) Example of the epidermis of a cotton stalk that failed to be extracted. (c) Example of the epidermis of broken cotton stalks.

Figure 13. Field test photo: 1. Reel wheel 2. V-shaped tooth roller 3. Torque transducer 4. Cotton stalks

Reviewer 5 Report

This reviewer thanks the authors' careful revisions, and recommends its publication.

This reviewer thanks the authors' careful revisions, and recommends its publication.

Author Response

Thank you for reviewers' valuable comments.